# A Novel 5-Chloro-*N*-phenyl-1H-indole-2-carboxamide Derivative as Brain-Type Glycogen Phosphorylase Inhibitor: Validation of Target PYGB

**DOI:** 10.3390/molecules28041697

**Published:** 2023-02-10

**Authors:** Yatao Huang, Shuai Li, Youde Wang, Zhiwei Yan, Yachun Guo, Liying Zhang

**Affiliations:** 1Laboratory of Traditional Chinese Medicine Research and Development of Hebei Province, Institute of Traditional Chinese Medicine, Chengde Medical University, Chengde 067000, China; 2Department of Pathogen Biology, Chengde Medical University, Chengde 067000, China

**Keywords:** PYGB knockdown, H/R injury, GP inhibitor, mouse astrocytes, oxidative phosphorylation

## Abstract

Brain-type glycogen phosphorylase (PYGB) inhibitors are recognized as prospective drugs for treating ischemic brain injury. We previously reported compound **1** as a novel glycogen phosphorylase inhibitor with brain-protective properties. In this study, we validated whether PYGB could be used as the therapeutic target for hypoxic-ischemic diseases and investigated whether compound **1** exerts a protective effect against astrocyte hypoxia/reoxygenation (H/R) injury by targeting PYGB. A gene-silencing strategy was initially applied to downregulate PYGB proteins in mouse astrocytes, which was followed by a series of cellular experiments with compound **1**. Next, we compared relevant indicators that could prove the protective effect of compound **1** on brain injury, finding that after PYGB knockdown, compound **1** could not obviously alleviate astrocytes H/R injury, as evidenced by cell viability, which was not significantly improved, and lactate dehydrogenase (LDH) leakage rate, intracellular glucose content, and post-ischemic reactive oxygen species (ROS) level, which were not remarkably reduced. At the same time, cellular energy metabolism did not improve, and the degree of extracellular acidification was not downregulated after administration of compound **1** after PYGB knockdown. In addition, it could neither significantly increase the level of mitochondrial aerobic energy metabolism nor inhibit the expression of apoptosis-associated proteins. The above results indicate that compound **1** could target PYGB to exert its protective effect against cellular H/R injury in mouse astrocytes. Simultaneously, we further demonstrated that PYGB could be an efficient therapeutic target for ischemic-hypoxic diseases. This study provides a new reference for further in-depth study of the action mechanism of the efficacy of compound **1**.

## 1. Introduction

Ischemic cerebrovascular disorders are major diseases that pose a serious threat to human health and life and whose yearly morbidity and mortality rates have steadily increased [1]. In periods of insufficient energy supply caused by brain ischemia, brain glycogen rapidly breakdowns to fulfill energy demand. Brain glycogen is mostly found in astrocytes, and it is the very important material basis for brain energy reserves and brain activity [2,3,4]. During brain ischemia, blood-borne glucose is insufficiently supplied, and astrocytic glycogen-derived lactate fuels the brain. Moreover, the glycogen breakdown during brain ischemia occurs at rates that are 200 times faster than the resting state [5]. Therefore, brain glycogen is rapidly depleted (within 4 min), and when this happens, the brain may suffer irreversible injury [6,7]. PYGB, which is mainly found in brain and heart tissues, is a key enzyme in brain glycogen breakdown and is also responsible for maintaining the optimal glycogen level for internal use [8,9,10]. Thus, PYGB has a crucial role in glycogen metabolism in the brain [11].

We previously revealed that compound **1** (Figure 1) is a novel glycogen phosphorylase inhibitor involved in glucose metabolism by inhibiting PYGB. Meanwhile, we also found that compound **1** has better inhibitory activity against PYGB (IC_50_ of 90.27 nM) and a potential therapeutic effect on brain ischemia [12,13]. In the present study, we initially applied a gene-silencing strategy to downregulate PYGB proteins and investigate the potential protective effect against astrocyte H/R injury with compound **1** by targeting PYGB. After PYGB knockdown, we performed a series of cytological experiments and confirmed that compound **1** showed its protective effect on astrocytes H/R injury by targeting PYGB and that PYGB could be used as an effective therapeutic target for ischemic-hypoxic diseases. The present study further verified the efficacy and potential therapeutic ability of compound **1**, providing a valuable reference for in-depth investigation of the pharmacodynamic action mechanism of compound **1**.

## 2. Results and Discussion

### 2.1. The Construction of Adeno-Associated Viral Vector to Knockdown PYGB Gene

In order to investigate whether there is a protective effect on mouse astrocytes with compound **1** after PYGB knockdown, we applied adeno-associated viral knockdown PYGB. A mouse PYGB knockdown adeno-associated viral vector (AAV2) was constructed to knock down PYGB proteins in mouse astrocytes. Then, the siRNA of target PYGB mRNA was designed using the online RNAi design software (Invitrogen), and shRNA-1/2/3 interference plasmids were constructed. Additionally, the downregulation efficiency of the AAV2-shPygb-1/2/3 interference plasmid was verified using real-time PCR (RT-PCR) assay after transfecting mouse astrocytes. Our results showed (Figure 2) that PYGB was successfully knocked down in mouse astrocytes, and the efficiency of AAV2-shPYGB-2 downregulation was significant.

### 2.2. After PYGB Knockdown, Compound ***1*** Lost the Protective Effect on H/R Injury in Mouse Astrocyte

After the successful isolation of mouse astrocytes [12], we established a H/R injury model and implemented it using an adeno-associated virus (AAV2) to further explore whether compound **1** could act against H/R injury in mouse astrocytes by targeting PYGB. Relevant studies have shown that the degree of H/R injury in mouse astrocytes could be reflected by cell viability, LDH leakage rate, intracellular glucose, and ROS level [14,15,16,17]. Therefore, we tested these indicators with their matching assay kits, as shown in Figure 3. First, compared with the blank group, the AAV2-negative control (NC) group and AAV2-shPYGB group showed no significant differences in the above indicators, indicating that virus-empty treatment (AAV2-NC) and virus intervention (AAV2-shPYGB) had no significant effect on cell viability, LDH, intracellular glucose, and ROS levels. Compared with the H/R + AAV2-NC group, cell viability was dramatically increased, LDH leakage rate, intracellular glucose, and ROS levels were effectively reduced in H/R + AAV2-NC + compound **1** (10 μM) group and in the H/R + AAV2-shPYGB group, which indicated that PYGB knockdown and inhibition of PYGB with compound **1** could both act against mouse astrocytes H/R injury. Moreover, after H/R treatment, the cell viability of the AAV2-shPYGB group was significantly reduced, and the LDH leakage rate, intracellular glucose, and ROS levels were significantly increased; however, these indicators did not significantly differ between the H/R + AAV2-shPYGB group and the H/R + AAV2-shPYGB + compound **1** group. This demonstrated that after the virus downregulation of PYGB proteins, cell viability could not be effectively enhanced by compound **1,** and LDH leakage rate, intracellular glucose, and ROS levels could not be significantly inhibited by compound **1**. These results indicated that PYGB knockdown and compound **1** inhibition of PYGB could both be used against mouse astrocytes H/R injury. Meanwhile, because virus intervention downregulated PYGB proteins, compound **1** failed to increase cell viability and showed no significant inhibitory effect on LDH leakage, intracellular glucose, and ROS levels. Thus, compound **1** could exert a protective effect on mouse astrocytes H/R injury by targeting PYGB.

### 2.3. After PYGB Knockdown, Compound ***1*** Could Not Improve Energy Metabolism in Mouse Astrocyte after Ischemia

Related studies revealed that glucose metabolism is closely related to ATP level [18,19,20]. In addition, ATP is an important energy molecule in cells, and the intracellular ATP content is used to evaluate the level of cellular energy metabolism [21,22,23,24]. Our previous report showed that compound **1** could dramatically reduce cellular ATP and remarkably improve energy metabolism in mouse astrocytes [12]. In this study, we wanted to determine ATP content further after PYGB knockdown to validate whether compound **1** worked by targeting PYGB. As shown in Figure 4, compared with the blank group, the AAV2-NC group and AAV2-shPYGB group showed no significant differences in cellular ATP levels, indicating that virus-empty treatment and virus intervention had no significant effect on ATP content. Moreover, the cellular ATP content of the above three groups with H/R treatment was obviously increased. Compared with H/R + AAV2-NC group, ATP content in groups H/R + AAV2-NC + compound **1** and H/R + AAV2-shPYGB was significantly reduced, thus indicating that PYGB knockdown and inhibition of PYGB with compound **1** both remarkably decreased cellular ATP content. In contrast to the H/R + AAV2-shPYGB group, the H/R + AAV2-shPYGB + compound **1** group showed no significant difference in reducing the level of cellular ATP, demonstrating that the level of cellular ATP content after hypoxia was not effectively reduced by compound **1** after the virus’ downregulating of PYGB proteins.

These results showed that after PYGB proteins were inhibited by virus intervention, compound **1** was ineffective in reducing ATP levels after mouse astrocyte H/R injury, which subsequently ceased to improve the level of intracellular energy metabolism. Thus, compound **1** could improve intracellular energy metabolism by targeting PYGB.

### 2.4. After PYGB Knockdown, Compound ***1*** Could Not Effectively Downregulate the Degree of Extracellular Acidification or Enhance Glycolytic Energy Metabolism in Mouse Astrocyte 

Previous studies have confirmed that lactate and pyruvate levels tend to increase due to enhanced anaerobic glycolysis during brain ischemia. In addition, lactate and its consequent acidosis have been identified as the leading causes of astrocyte injury after brain ischemia [25,26]. Evaluating the extracellular acidification rate (ECAR) is an indirect way to measure lactate secretion and glycolysis. Therefore, in order to investigate the influence of compound **1** on the extracellular acidification rate in mouse astrocytes after PYGB knockdown, we further determined the extracellular acidification rate (ECAR) using the Seahorse Cell Energy Analyzer. As shown in Figure 5, virus-empty treatment had no significant effect on the cellular glycolytic energy metabolism level, as there was no significant difference between the blank group and the AAV2-NC group for each indicator. However, the basal glycolysis level (glycolysis), glycolytic capacity (glycolytic capacity), and glycolytic reserve (glycolytic reserve) of these two groups of cells were significantly reduced after H/R treatment, demonstrating that cellular glycolytic energy metabolism was significantly inhibited. 

On the other hand, compared with the H/R + AAV2-NC group, the glycolytic capacity and glycolytic reserve of H/R + AAV2-NC + compound **1** and H/R + AAV2-shPYGB groups were significantly increased, indicating that PYGB knockdown and compound **1** inhibition of PYGB increased the glycolytic potential of astrocytes. Additionally, compared with the AAV2-shPYGB group, glycolytic capacity and glycolytic reserve of cells in the H/R + AAV2-shPYGB group were significantly increased; however, relative to the H/R + AAV2-shPYGB group, there were no significant differences in cells’ glycolytic capacity and glycolytic reserve after administration of compound **1**. Thus, after PYGB proteins were downregulated by the virus, the glycolytic potential was not obviously improved by compound **1**. Additionally, there was no significant difference in the level of non-glycolytic acidification.

The above results show that compound **1** significantly downregulated the degree of extracellular acidification and promoted cellular glycolytic potential and the levels of glycolytic energy metabolism. However, after the virus’ downregulating of PYGB proteins, the glycolytic potential of mouse astrocytes was not effectively enhanced by compound **1**. Therefore, compound **1** could improve glycolytic energy metabolism by targeting PYGB.

### 2.5. After PYGB Knockdown, Compound ***1*** Could Not Significantly Improve the Level of Mitochondrial Aerobic Energy Metabolism or Decrease Anaerobic Glycolysis

During the reperfusion, mitochondria generate a large amount of ROS, which could aggravate cellular and mitochondrial damage due to respiratory bursts. Thus, the cellular oxygen consumption rate (OCR) is commonly used to estimate the function of mitochondria [27,28,29]. Our previous report indicated that compound **1** could significantly improve the mitochondrial aerobic energy metabolism levels in astrocytes [12]. Therefore, in order to investigate the effect of compound **1** on the level of cellular mitochondrial aerobic energy metabolism after PYGB knockdown, we examined the cellular oxygen consumption rate with Seahorse XF.

As shown in Figure 6, compared with the blank group, there was no significant difference in groups AAV2-NC and AAV2-shPYGB in any of the indicators, showing that virus-empty treatment or PYGB knockdown had no obvious influence on the levels of mitochondrial aerobic energy metabolism in mouse astrocytes. However, after H/R treatment, the basal oxygen consumption capacity (basal respiration), ATP production, maximum oxygen consumption capacity (maximal respiration), and oxygen consumption potential (spare respiratory capacity) of cells belonging to the above three groups were significantly decreased, demonstrating that the levels of mitochondrial aerobic energy metabolism were remarkably inhibited.

In contrast to the H/R + AAV2-NC group, the level of cellular mitochondrial aerobic energy metabolism was significantly improved when cells were administered compound **1** or interfered with by the virus, as evidenced by increased cells’ basal oxygen consumption capacity (basal respiration), ATP production, maximum oxygen consumption capacity (maximal respiration), and oxygen consumption potential (spare respiratory capacity). Compared with the H/R + AAV2-shPYGB group, there were no significant differences in cells’ basal respiration, ATP production, maximal respiration, and spare respiratory capacity after administration of compound **1**. Therefore, after PYGB knockdown, the mitochondrial aerobic energy metabolism and anaerobic glycolysis levels were no longer significantly increased by compound **1**. Moreover, there was no significant difference between the levels of non-ATP synthesis oxygen consumption rate (proton leak) and non-mitochondrial oxygen consumption rate (non-mitochondrial respiration).

These results showed that PYGB knockdown and PYGB inhibited by compound **1** could significantly improve mitochondrial aerobic energy metabolism levels and reduce anaerobic glycolysis. Nonetheless, after the virus’ downregulating of PYGB proteins, compound **1** ceased to ameliorate mitochondrial energy metabolism. Therefore, compound **1** could improve cellular mitochondrial aerobic energy metabolism and decrease anaerobic glycolysis by targeting PYGB.

### 2.6. After PYGB Knockdown, Compound ***1*** Failed to Validly Inhibit the Expression of Apoptosis-Related Proteins

Apoptosis is one of the major routes leading to cell death and an essential process in cell H/R injury [30,31]. Bax protein has a pro-apoptotic role in cells, while caspase-3 is the most critical apoptosis-executing protease in apoptosis [32,33]. Therefore, inhibiting their expression could alleviate apoptosis. Moreover, our previous study indicated that compound **1** can inhibit the expression of apoptosis-related proteins Bax and caspase-3, thus alleviating apoptosis [12,13]. In this study, we further applied Western Blot to determine the expression of apoptosis-related proteins caspase-3 and Bax to verify whether the expression of apoptosis-related proteins was inhibited by compound **1** after PYGB knockdown. As shown in Figure 7, compared with the blank group, the expression of caspase-3 and Bax proteins did not significantly differ in the AAV2-NC group and AAV2-NC-shPYGB group, showing that virus-empty treatment and virus intervention had no significant effect on the expression of apoptosis-related proteins. After these three groups were treated with H/R, the expression of caspase-3 and Bax proteins was significantly increased; however, compared with the H/R + AAV2-NC group, the expression of caspase-3 and Bax proteins was distinctly restrained in the H/R + AAV2-NC + compound **1** group and the H/R + AAV2-shPYGB group, demonstrating that both PYGB knockdown and compound **1** inhibition of PYGB significantly reduced the expression of caspase-3 and Bax proteins. Additionally, compared with the H/R + AAV2-shPYGB group, the expression of caspase-3 and Bax proteins in the H/R + AAV2-shPYGB + compound **1** group was not significantly inhibited, which further indicated that the expression of caspase-3 and Bax proteins was not obviously inhibited by compound **1** after PYGB proteins knockdown.

The above results showed that PYGB knockdown and compound **1** dramatically decreased the expression of apoptosis-related proteins. However, after the virus interference in downregulating PYGB, the expression of apoptosis-related proteins was not effectively inhibited by compound **1**. Thus, compound **1** could inhibit the expression of apoptosis-related proteins by targeting PYGB, which, in turn, protects mouse astrocytes.

## 3. Materials and Methods

### 3.1. Animals

Male C57BL/6J mice were obtained from Beijing Huafukang Biotechnology Co., Ltd. (Beijing, China) within 24 h of birth. All animal studies (including the mice euthanasia procedure) were carried out in compliance with the regulations and guidelines of Chengde Medical University institutional animal care and conducted according to the Association for the Assessment and Accreditation of Laboratory Animal Care International (AAALAC) and the Institutional Animal Care and Use Committee (IACUC) guidelines.

### 3.2. Cell Isolation and Culture

C57 mice were used in the present study (24 h after birth). Euthanasia was performed by cervical dislocation [34], after which the mice were disinfected via immersion in 75% ethanol by volume after execution, and their meninges and vessels were removed. The brain (excluding the hippocampal portion) was placed in 1× PBS (pH 7.4) and rinsed to remove the superficial blood layer until the brain tissue became milky white. The tissue was shredded into crumbs with ophthalmic forceps, repeatedly cut with ophthalmic scissors for 10 min, and then gently blown with a pipette gun 20–30 times. Digestion termination solution was added at 1:1 to abort the enzyme reaction. The solution was then centrifuged for 10 min at 1000 rpm, after which the supernatant was then discarded, and the culture medium was added for resuspension. Then, cells were stained for counting. Cell density was adjusted to 1 × 10^6^ cells/mL, and they were inoculated into culture flasks. Primary mouse brain astrocytes were cultured in DMEM medium with 10% FBS, 1% penicillin–streptomycin in a CO_2_ incubator at 37 °C for 3 days.

### 3.3. Experiment Grouping

Cells were randomly divided into the eight following groups: blank group; H/R group; AAV2-NC; H/R + AAV2-NC group; H/R + AAV2-NC + compound **1** (10 μM) group; AAV2-shPYGB group; H/R + AAV2-shPYGB group; H/R + AAV2-shPYGB + compound **1** (10 μM) group. Negative control (NC): astrocytes were treated with the empty virus. Hypoxia/reoxygenation (H/R) treatment was as follows: astrocytes were maintained in a humidified incubator containing 95% air and 5% CO_2_ at 37 °C (called nonmonic conditions). Hypoxic conditions were achieved via exposure to a mixture of 95% N_2_ and 5% CO_2_ in a humidified incubator for 6 h, and reoxygenation was performed for an additional 24 h under nonmonic conditions.

### 3.4. PYGB Knockdown

Mouse PYGB knockdown AAV2 was constructed, packaged, and validated by Baiji Biomedical Technology (Shanghai, China) Co., Ltd.; Virus titer: 1 × 10^12^ VG/mL; Mouse PYGB gene: NCBI Reference Sequence: NM_153781.1.

### 3.5. CCK-8 Assay

Cell Counting Kit-8 was purchased from Tongren Chemical (Kumamoto, Japan). Cells were cultured in 96-well plates; 100 μL of mouse astrocyte suspension at a density of 1 × 10^4^ cells/mL was then prepared. The plates were precultured in an incubator (37 °C, 5% CO_2_) for 6 h. After the respective treatments (see cell grouping above), the plates were incubated in an incubator for 24 h, after which 10 μL of CCK-8 solution was added to each hole and incubated for another 1 h at 37 °C. The absorbance at 450 nm was determined using a microplate reader (Molecular Devices, San Francisco, CA, USA).

### 3.6. LDH Release

The LDH Assay Kit was purchased from Beyotime (Shanghai, China). According to the size and growth rate of the cells, an appropriate number of cells was inoculated into a 96-well plate so that the cell density did not exceed 80–90% when the cells were to be assayed. The culture solution was aspirated and washed once with PBS solution. After replacement with a new medium, the wells were treated (see cell grouping above), and culturing was continued as usual. One hour before the scheduled time point of the assay, the cell culture plate was removed from the incubator, and the cell lysate provided in the kit was added to the “sample maximum enzyme activity control wells” at 10% of the volume of the original culture medium. After adding the cell lysis solution, it was repeatedly blown and mixed several times, after which incubation in the cell incubator was continued. After reaching the predetermined time, the cell culture plate was centrifuged for 5 min at 400× *g* in a multi-well plate centrifuge. Then, 120 μL of supernatant from each well was taken and added to the corresponding well of a new 96-well plate, and the sample was then assayed.

### 3.7. Medium Glucose Content

After collecting the cell culture fluid, a standard curve was performed. Next, the assay detected intracellular glucose through grouping. More details are shown in our previous paper [12].

### 3.8. ROS Release

The DCF ROS Detection Kit was purchased from Thermo Fisher (Waltham, MA, USA). First, DCFH-DA (2,7-dichlorodihydrofluorescein diacetate) was diluted with a serum-free culture medium at 1:1000 to a final concentration of 10 µM/L. Cells were then collected and suspended in diluted DCFH-DA at a cell concentration of one to twenty million cells/mL and incubated for 20 min in a 37 °C cell incubator. Cells were mixed upside down every 3–5 min to bring the probe and cells into full contact. Cells were then washed three times with serum-free cell culture medium to adequately remove DCFH-DA that did not enter the cells. Assays were performed using flow cytometry (Thermo Fisher, Waltham, MA, USA).

### 3.9. ATP Content

ATP content was detected using an ATP assay kit (Beyotime, Shanghai, China). The culture fluid was aspirated, and the cells were lysed. For adequate lysis, repeated blowing or shaking of the culture plate was performed using a pipette to make full contact with the lysate and lyse the cells. After lysis, centrifugation at 4 °C 12,000× *g* for 5 min was performed, and the supernatant was removed for subsequent assays (centrifuge purchased from Eppendorf, Hamburg, Germany). A total of 100 µL of ATP assay working solution was added to the assay wells and left at room temperature for 3–5 min. Next, 20 µL of sample or standard was added to the assay wells and quickly mixed with a micropipette, after which the RLU value was measured with a luminometer after at least 2 s interval.

### 3.10. Mitochondrial Function and Cellular Metabolic Status

When using the Seahorse XF assay kit (Agilent Technologies, Palo Alto, CA, USA), cells were cultured on special microtiter plates, and the oxygen consumption rate (OCR) and the extracellular acidification rate (ECAR) were measured in real-time after the addition of different drugs to characterize the metabolic status of the cells. Detailed instructions for use can be found in Agilent’s Seahorse XF Kit Instructions [35].

### 3.11. Western Blotting

Western blotting was performed using standard methods. After homogenization and centrifugation, the total protein value of the supernatant was collected. The amount of protein was determined with the BCA protein assay kit. Fifty micrograms of protein from each sample were loaded onto sodium dodecyl sulfoxide polyacrylamide gel electrophoresis (SDS-PAGE), after which the membrane was transferred to a plate containing TBST solution, destained at room temperature, and blocked by slowly shaking on a shaker for 2 h. Antibody reactions were performed after blocking nonspecific binding sites with 5% bovine serum albumin. Blocked membranes were incubated with the primary antibodies (Bax Antibody and caspase-3 (D3R6Y) Rabbit mAb, Cell Signaling Technology, Danvers, MA, USA) overnight at 4 °C, after which the membrane was washed with TBST and incubated with a secondary antibody conjugated with horseradish peroxidase. After three washes, proteins were visualized through enhanced chemiluminescence detection. Blots were detected using a Gel Doc 2000 (Bio-Rad, From Hercules, CA, USA).

### 3.12. Statistical Analysis

Data were expressed as the mean ± SD. The measured variables between the experimental and control groups were assessed using Student’s t-test for nonparametric data. *p* < 0.05 represented statistical significance.

## 4. Conclusions

Cell injury caused by brain ischemia is an ongoing issue. We previously reported that compound **1** could alleviate H/R injury in mouse astrocytes and has a potential therapeutic effect on brain ischemia. Herein, we knocked down PYGB proteins using AAV2 first and established a H/R injury model in order to investigate whether compound **1** could exert protective effects on brain injury by targeting PYGB. Subsequently, we evaluated the effects of compound **1** on cell viability, LDH leakage rate, intracellular glucose content, post-hypoxic ROS levels, post-hypoxic ATP levels, cellular oxygen consumption rate, degree of extracellular acidification, and apoptosis-related protein expression after PYGB knockdown. Our results showed that after PYGB knockdown, compound **1** could not significantly alleviate astrocytes’ H/R injury as evidenced by cell viability, which failed to show significant improvement, and LDH leakage rate, intracellular glucose content, and post-ischemic ROS level, which were not remarkably reduced. Meanwhile, by administering compound **1** after PYGB knockdown, we found that cellular energy metabolism was not significantly improved, and the degree of extracellular acidification was not remarkably downregulated. Additionally, it could neither increase the level of mitochondrial aerobic energy metabolism nor inhibit the expression of apoptosis-associated proteins.

Taken together, these results indicate that compound **1** can alleviate cellular H/R injury by targeting PYGB and verified PYGB as the effective therapeutic target for ischemic-hypoxic diseases. At the same time, these results further validated the potential therapeutic ability of compound **1**. Therefore, the present study provides a valuable reference for further in-depth study of the efficacy action mechanism of compound **1**, as well as novel insights for further development of the PYGB target.

## Figures and Tables

**Figure 1 molecules-28-01697-f001:**
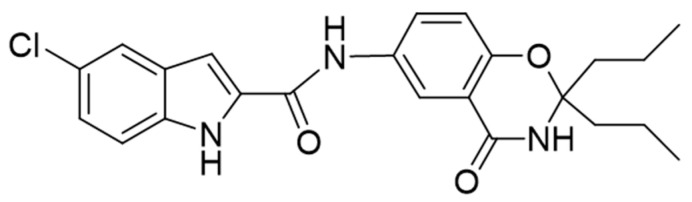
The structure of compound **1.**

**Figure 2 molecules-28-01697-f002:**
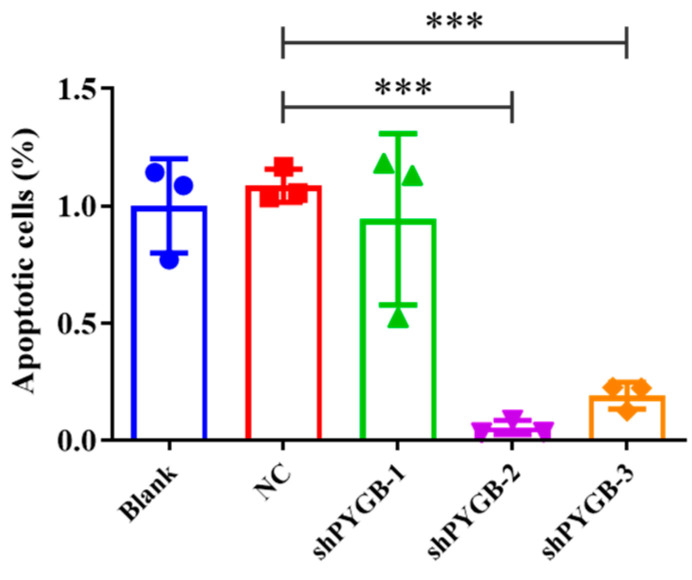
RTPCR for screening adeno-associated viruses. Data represent mean ± SD (*** *p* < 0.001).

**Figure 3 molecules-28-01697-f003:**
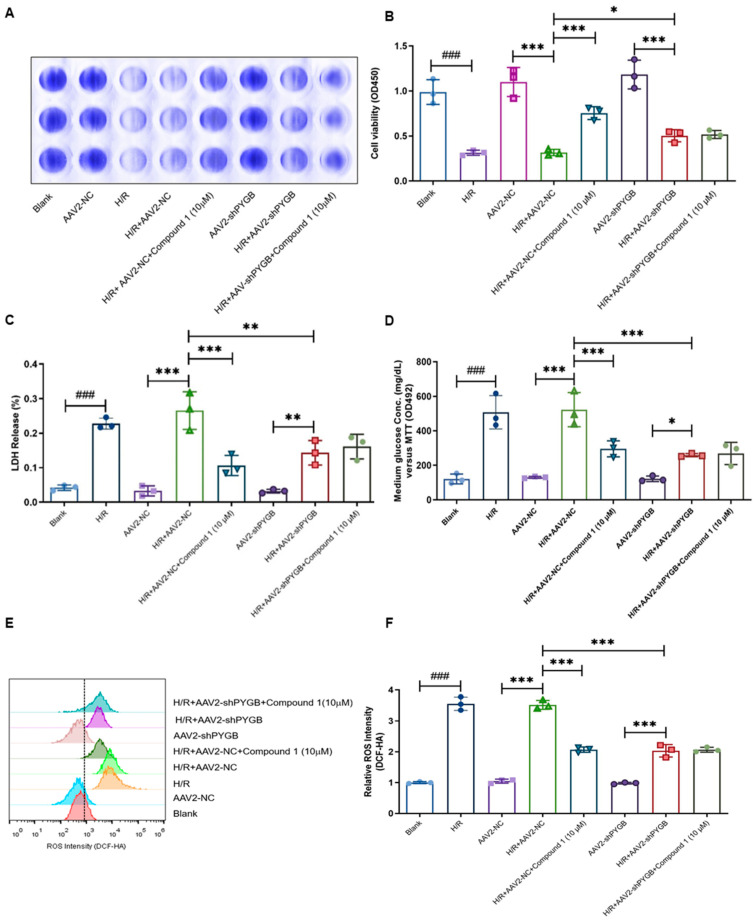
(**A**,**B**) Effect of compound **1** on cell viability in astrocytes H/R injury after PYGB knockdown. (**C**) Impact of compound **1** on LDH release in astrocytes H/R injury after PYGB knockdown. (**D**) Influence of compound **1** on intracellular glucose consumption in astrocytes H/R injury after PYGB knockdown. (**E**,**F**) Effect of compound **1** on ROS production induction by astrocytes H/R injury after PYGB knockdown. Data represent mean ± SD (* *p* < 0.05, ** *p* < 0.01, *** *p* < 0.001, ^###^ *p* < 0.001).

**Figure 4 molecules-28-01697-f004:**
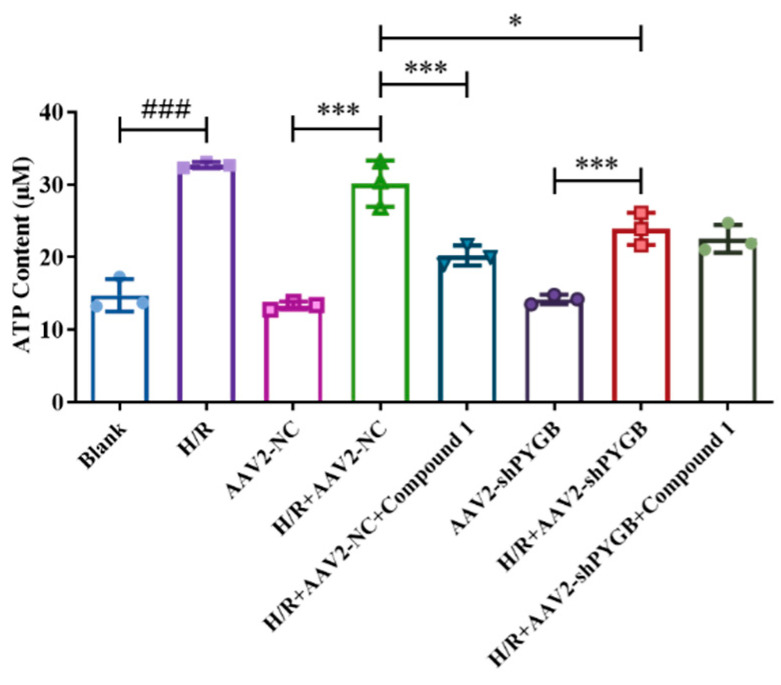
Effect of compound **1** on post-ischemic intracellular ATP content after PYGB knockdown. Data represent mean ± SD (* *p* < 0.05 *** *p* < 0.001; ^###^ *p* < 0.001).

**Figure 5 molecules-28-01697-f005:**
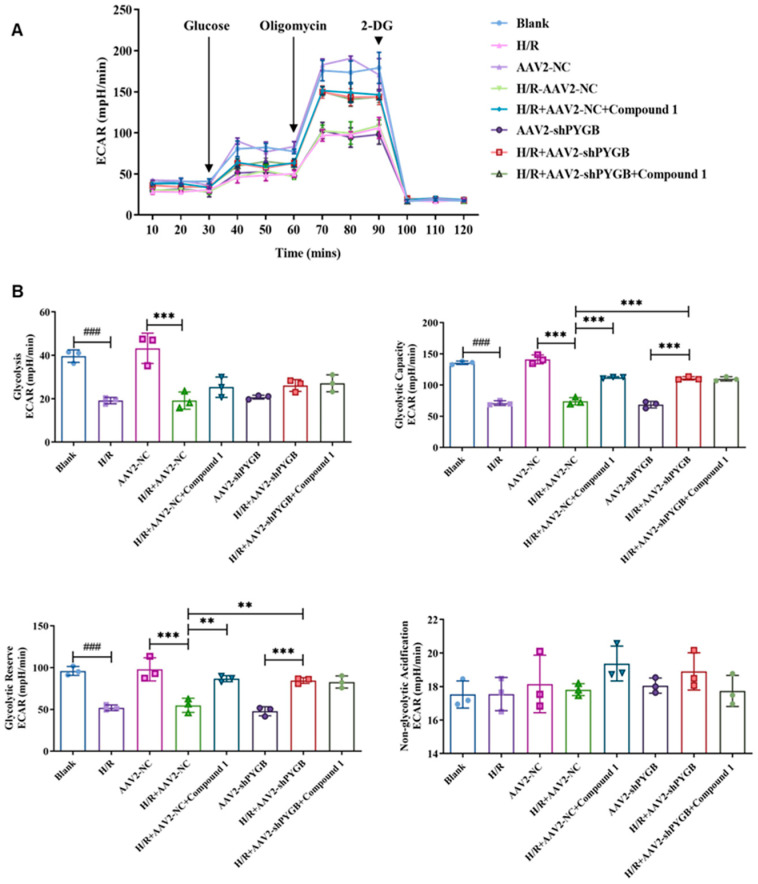
(**A**) Mouse astrocytes after H/R treatment with various drug administration methods; ECAR was detected using the Seahorse XF Extracellular Flux Analyzer. Three drugs were added sequentially: glucose, oligomycin, and 2-DG. (**B**) Bar graphs of glycolysis, glycolytic capacity, glycolytic reserve, non-glycolytic acidification. Data represent mean ± SD (** *p* < 0.01 *** *p* < 0.001; ^###^ *p* < 0.001).

**Figure 6 molecules-28-01697-f006:**
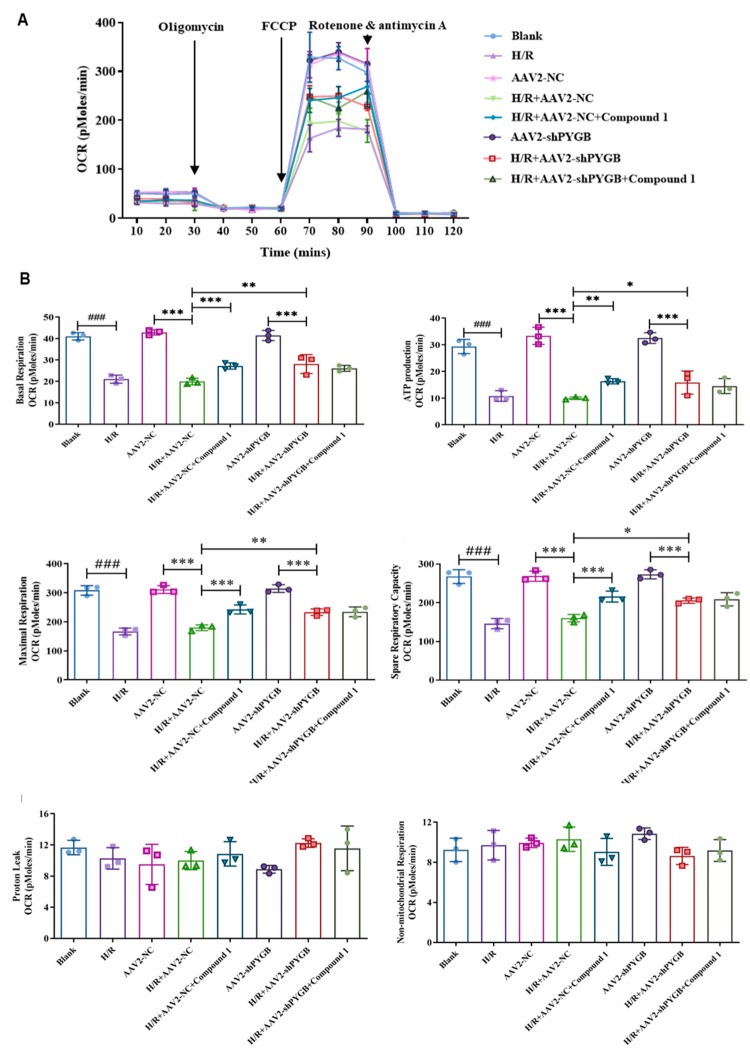
(**A**) Mouse astrocytes after H/R treatment with various drug administration methods; the real-time OCR was measured using Seahorse XF96 Extracellular Flux analyzer. The basal OCR was measured at three time points, after which four chemicals were sequentially injected into the medium: the ATP synthase inhibitor oligomycin, the uncoupler FCCP, rotenone, and antimycin A. (**B**) Bar graphs of basal respiration, ATP production, maximal respiration, spare respiratory capacity, proton leak, and non-mitochondrial respiration. Data represent mean ± SD (* *p* < 0.05, ** *p* < 0.01 *** *p* < 0.001; ^###^ *p* < 0.001).

**Figure 7 molecules-28-01697-f007:**
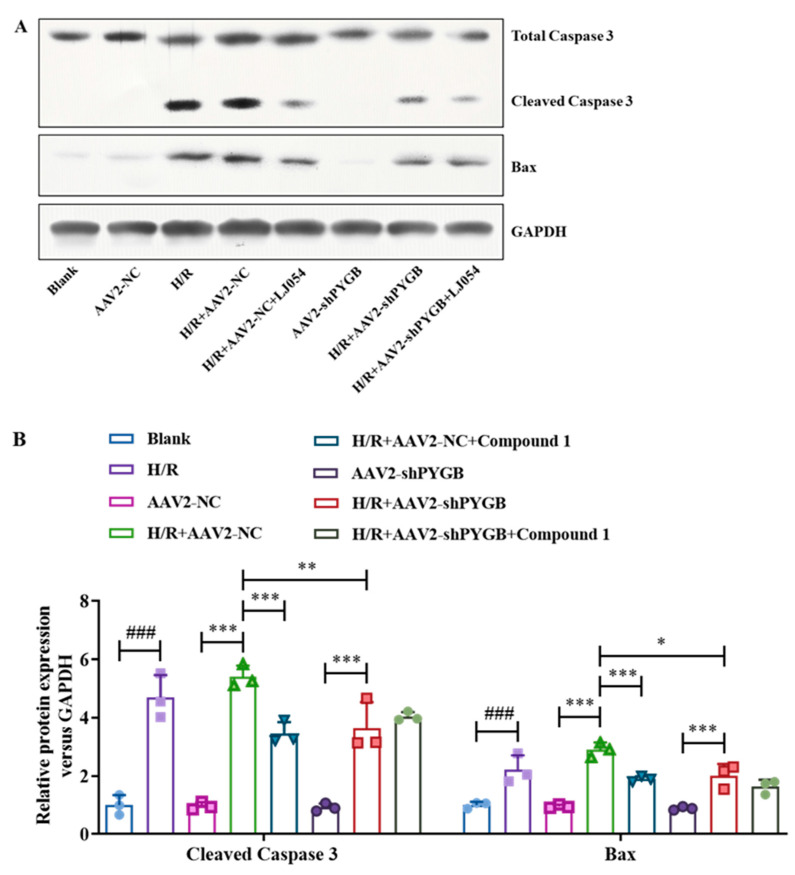
(**A**) Western blot image of total caspase-3, cleaved caspase-3, Bax, GAPDH protein. (**B**) Bax and cleaved caspase-3 protein expression. Protein content was normalized to GAPDH. Data represent mean ± SD (* *p* < 0.05, ** *p* < 0.01, *** *p* < 0.001; ^###^ *p* < 0.001).

## Data Availability

The data presented in this study are available on request from the corresponding author.

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
