# Peer review of "A Novel 5-Chloro-N-phenyl-1H-indole-2-carboxamide Derivative as Brain-Type Glycogen Phosphorylase Inhibitor: Validation of Target PYGB"

_molecules, 2023, doi:10.3390/molecules28041697_

Round 1

Reviewer 1 Report

The authors report the evaluation of the compound 1 activity in the management of the Hypoxia/reperfution injury. The goodness of the compound was tested in astrocytes cells in which the PYGB was silenced. 

Despite the general quality of the manuscript the major concern is about the novelty of the data. The reported results are very close to the data reported in a previous article of the same research unit. (Molecule2022 Sep 26;27(19):6333.doi: 10.3390/molecules27196333.) The new results did not add new information about the molecule, its mechanism of action and its therapeutic effect. Thus, I do not suggest the manuscript for publication.

Reviewer 2 Report

This is an interesting study, may be accepted for the publication after major revision. 

1-Culture: Primary mouse brain astrocytes were cultured using DMEM medium with 295 10% FBS, 1% Penicillin-Streptomycin, etc. in a 37°C, CO2 incubator. How many cells were initially seeded, give the number of cells/ml and how long they culture the cells, indicate the number of days ?

2- After the mice (C57) were sacrificed- How did they sacrifice the animals, they need to provide the method

3- Western blot- Which primary antibodies they used it, and describe this method in detail.

4- What is the relationship between Bax and Cleaved Caspase 3 protein expression after the treatment and how they regulate the cell growth and proliferation.

5-It would be interesting to see impact of 5-Chloro-N-phenyl-1H-indole-2-carboxamide Derivative as Brain-Type Glycogen Phosphorylase Inhibitor in brain cancer cells

6-They must do MTT assay to show the cell viability after treatment 5-Chloro-N-phenyl-1H-indole-2-carboxamide Derivative

7-There are many grammatical errors which need to be fixed. 

Reviewer 3 Report

I had a great pleasure reading this well-written manuscript. Although the chemistry content of this work is quite low compared to the amount of biology work reported (not typical for this chemistry-oriented MDPI journal), I believe the readership of Molecules journal will benefit greatly from disclosure of these important results. Apart from a few language idiosyncrasies detected here and there thoughout the article text, the submission offers next to nothing in terms of grounds for criticism. This work should be accepted and publushed as is in Molecules journal. 

Round 2

Reviewer 1 Report

The authors updated the manuscript, the English was fully revised and ameliorate. The authors stated that the reported results are the continuation  of already published data about the studied compound. The experiments are well planned and described. 

Reviewer 2 Report

The revised manuscript is improved and may be accepted for the publication